# Preparation and Microstructure of Alkali-Activated Rice Husk Ash-Granulated Blast Furnace Slag Tailing Composite Cemented Paste Backfill

**DOI:** 10.3390/ma15134397

**Published:** 2022-06-22

**Authors:** Wenhua Zhao, Ceyao Ji, Qi Sun, Qi Gu

**Affiliations:** 1College of Architecture and Transportation, Liaoning Technical University, Fuxin 123000, China; zhwh09@163.com (W.Z.); ceyao_ji@163.com (C.J.); 2School of Civil Engineering, Liaoning Technical University, Fuxin 123000, China; m19824853308@163.com

**Keywords:** alkali excitation rice husk ash, GGBS composite cementitious material, strength formation mechanism, microstructure, filling

## Abstract

At present, the research on rice hull ash and cement-based materials as cementitious materials continues to deepen. Low-cost rice hull ash replaces part of Portland cement, which plays a dual role in saving material costs and improving environmental benefits. In this study, alkali-activated rice husk ash (RHA) and ground granulated blast furnace slag (GGBS) were used to prepare cementitious material. The influence of RHA dosage on the strength, slump degree, and coagulation time of cementitious material was studied. On this basis, tailing was used as an aggregate based on the orthogonal design method and the bone–gel ratio, modulus, and alkali content were taken as variable factors, with strength and slump degree taken as the targets. A new cemented paste backfill (CPB) was prepared and mix ratio optimization was carried out. The strength formation mechanism of cementitious material and CPB was explored by combining scanning electron microscopy (SEM), energy dispersive spectrometry (EDS) and *X*-ray diffraction (XRD). The results of this study show that with the increase in RHA mixing, the unconfined compressive strength (UCS) of the gelling material purification slurry showed a tendency first to increase and then decrease. When the amount of RHA was about 10%, the internal structure became denser, more C-S-H gel was generated, and greater strength could be obtained. The specific surface area of RHA is high, and a small amount of RHA can fill the internal pores, making the internal structure of concrete more dense. The active silica content in RHA is relatively high. The addition of RHA can appropriately improve the strength of the material, which is of certain significance to our material research. Finally, the micro-analysis of RHA-GGBS clean slurry, the analysis of influencing factors of fluidity and strength, and the optimal mix proportion of alkali-activated RHA-GGBS-based backfill are put forward.

## 1. Introduction

The mining of mineral resources is often accompanied by a large amount of waste accumulation, which not only covers a large area, but also pollutes the environment. It is of certain research significance to make full use of these wastes to prepare a CPB. The paste filling method is widely used to solve the problems of uneven settlement and collapse in goaf. Use of paste backfill mining has been increasing in recent years, but the vast majority of CPB uses ordinary Portland cement (OPC) as a cementitious material. OPC produces a large amount of carbon dioxide in the production process, and the cost is relatively high [1,2,3]. Rice husk, a common agricultural solid waste, when calcined into RHA has a certain volcanic ash activity [4]. If a new type of CPB can be prepared by using alkali-activated RHA and GGBS as cementitious material, with tailings as aggregates, the problem of carbon dioxide emissions and high costs in the production of OPC can be solved. RHA and GGBS are both pozzolanic active materials. The two materials are used as cementitious materials, activated by alkali, and with tailings as aggregate to prepare a new type of backfill, which has certain research significance.

In the preparation of gelling materials from RHA, Salas J. et al. [5] found that RHA is rich in silica and, with appropriate combustion techniques, can be produced from rice husks for use in concrete as a supplementary cementitious material (SCM). Thomas B S et al. [6] showed that RHA has the advantages of high strength, low shrinkage, low permeability, high carbonation resistance, chloride resistance, sulfate resistance, and an acidic environment resistance. De Sensale G R. [7] studied different proportions of RHA levels, two RHAs (amorphous and partial crystalline states optimized by dry grinding), and several hydro-glue ratios. Park K-B et al. [8] showed that the calcium hydroxide content in the cement–RHA mixture decreases with the increase in the rate of RHA substitution. The proposed hydration model was verified by the test data of RHA–blended concrete with different water-to-glue ratios and different RHA replacement rates. Memon M J et al. [9] found that concrete with 15% RHA and 0.25% polypropylene (PP) had better performance than concrete without SCM. Bheel N et al. [10] used RHA as a partial alternative to cement in concrete to reduce its costs, and alternative treatment methods using agricultural/industrial waste were discovered. The main objective was to determine the properties of freshly mixed (flowability) and hardened (split tensile strength and compressive strength) concrete using RHAs of 0%, 5%, 10%, 15%, and 20% (by weight). The study of Siddika A et al. [11] found that RHA concrete is more resistant to chloride-ion permeation than ordinary Portland cement concrete. Nasiru S et al. [12] studied the volcanic ash reaction and microfilling effect induced by RHA and found that the addition of RHA improved the mechanical properties and durability of cement mortar at a later curing age. Safiuddin M et al. [13] used RHA instead of 0–30% cement (by weight). The fresh properties studied included fillability, passability, segregation resistance, air content, and unit weight. The effects of RHA and *W/B* ratios on these performances were observed. The experimental results showed that the water–ash ratio and RHA content have a significant impact on the mixing performance of concrete. RHA also affects the introduction of air, reducing the unit weight of the concrete. The study by Antiohos S K et al. [14] found that RHA is extremely “sensitive” to changes in fineness; the higher the fineness, the more aggressive the effect of RHA in the mixture. Unsurprisingly, active silica plays a key role in later strength increases, suggesting that as hydration develops, the ash effect replaces the “physical” effect of ash. De Sensale G R. [15] found that residual RHA had a positive effect on early compressive strength, but the long-term performance of RHA concrete that controlled incineration was more significant. The results of split tensile resistance and air permeability reveal the importance of filler and ash effects for concrete containing residual RHAs and RHA produced by controlled incineration. The study by Bie R S et al. [16] found that RHA obtained under the right conditions could be used as a cement additive to improve the compressive and flexural strength of cement mortar specimens. The results showed that an RHA rate of 10% (by weight) has the best effect concerning enhancing cement strength. A study by Chao-Lung H et al. [17] obtained compressive strengths of cylindrical concrete in the range of 47–66 MPa. The results also showed that up to 20% of ground RHA could be mixed with cement without adversely affecting the strength and durability of the concrete. Sakr K. [18] found that RHA-doped concrete has good sulfate resistance, while silica fume (SF) doped concrete has better sulfate resistance. SF or RHA has no significant effect on the gamma attenuation coefficient of concrete. The results showed that concrete with RHA has better mechanical and physical properties than concrete without additives, but its performance is lower than concrete with SF added.

In the development of novel CPB, Fall M et al. [19] showed through their results that the absorption of sulfates by calcium silicate hydrate (C-S-H) might lead to the formation of lower mass C-S-H, thereby reducing the strength of CPB. Koohestani B et al. [20] studied and compared the effects of non-polar organosilanes (vinyl and methyl) and high-efficiency polycarboxylate superplasticizers on the flow behavior, strength development, and microstructural properties of CPB composed of vulcanized and non-vulcanized tailings. The results showed that the use of vinyl silanes was more effective in densifying the CPB matrix due to improved hydration and the formation of additional C-S-H gels in non-vulcanized CPB. Chen S et al. [21] found that the fundamental factor affecting GCPB strength was the relationship between the volume of the void and the amount of calcium silicate gel, calcium alum, and Fridel salt formed. Mangane M B C et al. [22] showed that the effect of highly effective water reducers on CPB performance depends on the type and dosage of the admixture. Polycarboxylates exhibit the best performance and allow the target consistency to be achieved at a lower moisture content (6% to 10%) without altering the mechanical strength of the CPB. Fall M et al. [23] studied the stress–strain characteristics of CPB under uniaxial compression and conventional triaxial testing. The results showed that the constraints, the age, and the intensity of CPB and their composition have a great influence on the stress–strain behavior of CPB. Increased confining pressure leads to changes in destruction patterns, stiffness, and strength. Wang Z et al. [24] conducted strength tests on new paste filler materials using coal zircon, laterite, and cement as materials. The experimental results showed that the optimal ratio of the new CPB in the coal mine is a 6:2:1 quality ratio of gangue, laterite, and cement, and the slurry concentration is 80%, which not only meets the transportation requirements of the filling process, but also reduces the filling cost. Yan B et al. [25] found that to improve the working performance of CPB and the efficiency of field engineering, mixed hydrophobic agents incorporated into CPB mud enhance the dewatering efficiency. Khaldoun et al. [26] introduced the characterization and formulation validation of a waste-priced solution in operation using CPB techniques. Paste backfilling (PBF) technology, due to its multi-purpose nature, results in increased resource recovery, suitable for most mining methods. The content and characteristics of each component of PBF (tailing sand, cementitious material, and water) directly affect the mechanical strength of the filling after hardening. The slurry density is the decisive factor in the strength of the cemented filler, and the combination of cementitious material chemistry and mixed water chemistry affects the formation of primary and secondary hydrates. Panchal et al. [27] elaborated on the research progress of using a dolomite limestone uranium deposit, which comprises large angles and fine particle sizes, to process the tailings of CPB. This method was found to be suitable for CPB and displayed good shear resistance. The CPB had sufficient strength to provide support for a column, roof, and wall. When examining the engineering and physical properties of the wet smelting tailings of a carbonate matrix uranium ore, the hydration time was positively correlated with the shear rate and yield strength, the rheological characteristics of the CPB were sensitive to the moisture content, and the slump degree changed significantly. Chen Q et al. [28] used phosphogypsum and phosphate tailing sand as aggregates in different combinations. When using cement or GGBS as a cementing agent and CaO as an admixture, when the cementitious material is GGBS, the compressive strength can be increased by three times, but the strength decreases after 28 days. However, when the cementitious material applied is cement, no gas is generated, the strength is improved after 28 days, and it can be used as a filling material. Lu H et al. [29] found that a large amount of tailings had accumulated in an open pit that could not be treated. The tailings were used as filling materials with the addition of paste filling, which solved the problem of tailing accumulation and enabled the full use of resources. Lang L et al. [30] studied a predictive model for tailing sand-cemented fillers based on the relationship between major component content and rheological properties. The prediction accuracy was significantly improved by combining the BP neural network with the principal component analysis (PCA) method compared with the BP neural network alone. Kesimal A et al. [31] described the effects of the physical, chemical, and mineralogical properties of tailings and cementitious materials using two different vulcanized tailings (tailings T1 and T2) and Portland cementitious volcanic ash materials (B1 and B2) on the short- and long-term lateral UCS of CPB samples. The results showed that the short-term strength development of the paste backfill sample inherently depends on the properties of the tailings and cementitious material used. As the water–ash ratio decreases, the short-term strength of CPB samples generally shows an upward trend. Chen Q et al. [32] examined the feasibility of recycling two different solid wastes, phosphogypsum (PG) and construction demolition waste (CDW), as CPB. The environmental impact of PG and CDW-based CPB was studied through a static leaching test, a rotary acid leaching procedure, and index detection. A new backfill system and process were also developed for engineering applications. The results show that the technology is a reliable and environmentally friendly alternative to recovering PG and CDW while supporting safe mining. Li X et al. [33] demonstrated the feasibility of PG-based CPB in terms of both physical properties and environmental effects, thus providing an environmentally friendly method through which to treat PG. The results showed that the release of P, F, and metals was significantly reduced after the addition of PG to the backfilled samples.

The aforementioned studies conducted in depth research on RHA cementitious material and new CPB, but there are few studies on the preparation of CPB using alkali-activated RHA and GGBS as the cementitious material, which requires further study.

In this paper, a new CPB is prepared with water glass and NaOH as alkali excitors, RHA and GGBS as cementitious materials, and tailing sand as aggregates The mix ratio of CPB is optimized, and the microstructure and formation mechanism of SEM and EDS are used to analyze the microstructure and formation mechanism. On the one hand, our research involved in depth research on the RHA-GGBS cemented paste backfill. Through microscopic analysis and fluidity influence analysis, it is proposed that RHA plays a key role in the compressive strength of clean slurry. On the other hand, the filling preparation and mechanical properties of RHA-GGBS cemented paste backfill were studied. Through orthogonal test, the optimum mix proportion of alkali-activated RHA-based backfill material was determined.

## 2. Experiment

### 2.1. Raw Materials

The main cementitious materials in this study are S95 GGBS micronized powder, produced by Shandong Kangjing New Material Technology Co., Ltd. (Jiaozhou, China); the product is gray white. The high fineness and high activity powder obtained from water-quenched blast furnace slag after drying, grinding, and other processes are Cao, SiO_2_, Al_2_O_3_, and RHA, produced in Hubei; the aggregate is tailing, which was obtained from Fuxin Mongol Autonomous County Source Phosphorus Mining Co., Ltd. (Fuxin, China). The particle size distribution of RHA, GGBS, and tailing was determined by a laser particle size distribution instrument, as shown in Figure 1. The chemical composition of RHA, GGBS, and tailing was determined by *X*-ray fluorescence spectroscopy, and the specific chemical composition is shown in Table 1. The mineral composition of the tailing was determined by *X*-ray diffraction. According to the XRD spectrum shown in Figure 2, the mineral composition of the tailing is mainly quartz, sodium feldspar, and diolite. The alkaline catalyst used to analyze pure solid particles in this study was commercially available sodium hydroxide with a purity of 96%, purchased from Liaoning Quanrui Reagent Factory, and the water glass used was produced by the Jiashan Yourui Refractory Materials Company.

### 2.2. Method

The mix ratios used in this study are shown in Table 2. The proportion of RHA to cementitious material was 0%, 5%, 10%, and 15% for corresponding groups RHA1-RHA4, respectively. After many attempts in the early stage, the experimental parameters of the water–glue ratio of 0.35, alkali excitant modulus of 1.0, and alkali used were 4% of the quality of cementitious material. The mixing water used in the experiment was divided into two parts—one for dissolving the solid particles of sodium hydroxide and the aqueous glass solution and the other for rinsing the solution remaining in the glass beaker during the pouring of the solution, which was stirred until it was homogeneous, and was left at room temperature for 10 min as a standby treatment. After weighing the gelling material used in the experiment, the cementitious material and the solution were placed in the mixer and stirred for 2 min to ensure the homogeneity of the material in the mixer. The fresh mixture was poured into a 40 mm diameter × 40 mm × 40 mm mold; three specimens were prepared for each age to test the UCS, and all the mixtures, after being poured into the mold, were placed in a shaker for 40 s. The mold containing the mixture was placed in a standard concrete maintenance box at a temperature of 20 ± 1 °C and humidity of ≥96% (maintenance box model YH-40B). The specimen was removed after one day of curing and stored in the above environment. Uniaxial compression, SEM, and EDS experiments were carried out after 28 days of curing. Uniaxial compression experiments were carried out using universal testing machines, and SEM and EDS analyses were performed using scanning electron microscopes.

## 3. Results and Discussion

### 3.1. Analysis of Experimental Results of Alkali-Activated RHA-GGBS Cementitious Material

#### 3.1.1. Flow Degree Slump Test Results

The test results of the RHA1 group–RHA4 group are: 295 mm, 285 mm, 272 mm, and 245 mm. The flowability slump test results are shown in Figure 3. When the first group of RHA is 0%, the cementitious material is all GGBS, the fluidity is the largest, and the flow degree slump of RHA1 reaches 29.5 cm. When the RHA replaces the GGBS by 15%, reaching the maximum value of the substituted GGBS, the flow degree slump of RHA4 is 24.5 cm, which is the minimum flow degree slump. For RHA4, compared with RHA1, the flow degree slump is reduced by 16.9%; for RHA3 vs. RHA1, the flow degree slump is reduced by 7.8%; finally, for RHA2 vs. RHA1, the fluidity slump decreases by 3.4%.

**Figure 3 materials-15-04397-f003:**
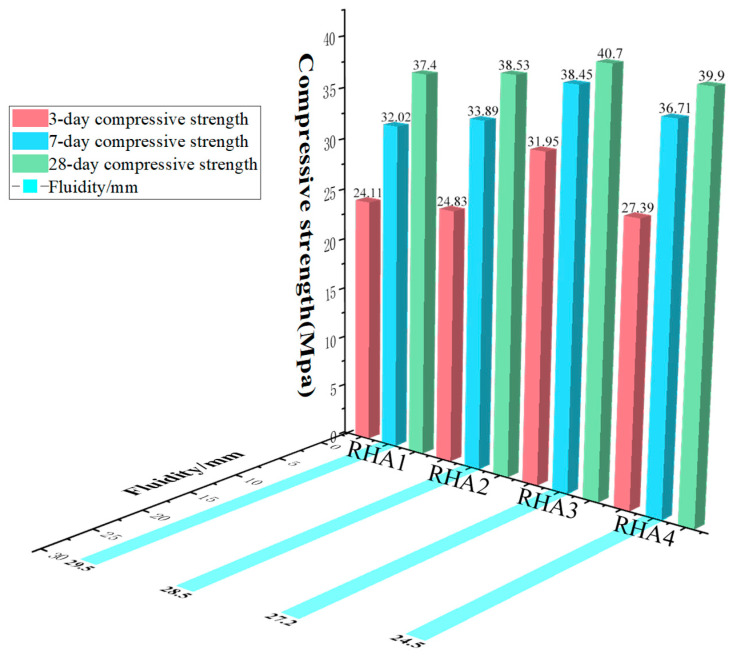
Fluidity diagram and compressive strength diagram of clean slurry.

RHA absorbs water more readily than GGBS, which is related to the structure of RHA, which is composed of a large number of porous, non-dense particles with an interior that contains a large number of pores. With the increase in the amount of RHA, the fluidity becomes smaller and smaller, and the finely ground RHA has good water absorption [34].

#### 3.1.2. Compressive Strength Test Results

The compressive strength test results of the specimens at different maintenance ages are shown in Figure 3. From the perspective of compressive strength, the compressive strength of the RHA group showed an increasing trend compared with the control group. The test results show that when the group was RHA3, that is, when GGBS and RHA accounted for 90% and 10% of the cementitious material, the compressive strength at each age was better than that of the other groups [35]. After the optimal amount of RHA was reached, the intensity showed a downward trend as the amount of RHA continued to increase.

(1)The influence of RHA on the strength of the net pulp

The alkali excitation GGBS purification group was compared with the different RHA-mixed pure pulp groups, and RHA3 had the highest strength. At 3 days, RHA1 intensity was 24.11 MPa. RHA3 reached 31.95 MPa, an increase of 32.5% compared to the baseline group RHA1. At 28 days, RHA1 intensity was 37.4 MPa, and RHA3 intensity was 40.7 MPa, an 8.8% improvement over the baseline group RHA1. The results show that the intensity of RHA on alkali excitation GGBS was improved to a certain extent [36], as the larger the specific surface area of RHA, the stronger the ability to adsorb Ca(OH)_2_, and the size of the specific surface area had a certain impact on the amorphous SiO_2_ volcanic ash activity. The active SiO_2_ reacts with Ca(OH)_2_ to form a C-S-H gel that enhances the adhesion between the structures, leading to a denser structure.

Compared to RHA1, the intensity of RHA4 increased by 13.6% at 3 days and 6.8% at 28 days. Compared to RHA3, RHA4 experienced a 14.3% decrease in intensity at 3 days and 1.8% at 28 days. When the amount of RHA was too high, the amount of additional silica was too large, and the reaction cannot be fully completed; therefore, the SiO_2_ activity is not fully excited, and the polymerization rate is slow. When there is too much RHA, the resulting micropores and fissures affect the strength of the net pulp.

(2)The growth trend of compressive strength in aging

In the alkali-activated RHA–GGBS clean pulp test block, with an increase in age, the compressive strength continued to increase. At 7 days, the RHA1 intensity was 32.02 MPa, reaching 85.6% of the 28-day intensity; RHA2 intensity was 33.89 MPa, reaching 87.9% of the 28-day intensity; RHA3 intensity was 38.45 MPa, reaching 95% of the 28-day intensity; and RHA4 intensity was 36.71 MPa, reaching 91.9% of the 28-day intensity. The slope of the early growth is steeper, and the slope of the later period is slightly gentler, which indicates that the strength growth rate of the net pulp test block within 7 days was faster, while the growth rate up to 28 days was somewhat slower.

(3)The effect of alkali excitors on compressive strength

Due to the disintegration of the GGBS vitreous network, it is easy to destroy the Ca-O, Si-O, and Al-O bonds of the GGBS particles and dissolve the silicate ions and aluminate ions, which can react with calcium ions to generate a hydrated calcium silicate gel, which makes the early structure dense and improves its strength.

#### 3.1.3. Condensation Time

When the water glass modulus is 1.0 and the alkali content is 4%, the initial coagulation time of the alkali-activated RHA-GGBS purification slurry is 31–35 min, and the final coagulation time is 43–57 min. When the amount of RHA is 0%, the initial and final coagulation times are 36 min and 57 min, respectively; when the RHA is 5%, the initial and final coagulation times are 35 min and 51 min, respectively; when the RHA is 10%, the initial and final coagulation times are 35 min and 46 min, respectively; when the RHS is 15%, the initial and final coagulation times are 31 min and 43 min, respectively. Judging from the experimental results, as the amount of RHA blending grows, the impact on the initial coagulation time is small, and the final coagulation time is shortened. The results of the condensation time tests of different groups for specimens are shown in Figure 4.

#### 3.1.4. Micro-Analysis

(1)XRD analysis

The XRD analysis of alkali-activated GGBS net slurries with different RHA incorporation amounts at 3 days and 28 days is shown in the Figure 5.

It can be seen from the figure that the hydration products of alkali-activated GGBS are mainly C-S-H gel, calcite, and clinoptilolite regardless of the change of the content of RHA, and do not change significantly with the extension of hydration time. Among them, the diffraction peaks of C-S-H and calcite overlap. Calcite is caused by dissolved Ca^2+^ in GGBS and atmospheric carbonation [37]. The change of diffraction intensity in the spectrum is mainly caused by the amount of RHA. The addition of RHA can promote the formation of more hydration products. At the same time, RHA3 can promote the formation of C-S-H and clinoptilolite more than RHA1, improving the compressive strength of alkali-activated GGBS paste. Because RHA produces a large amount of SiO_2_, SiO_2_ and Ca(OH)_2_ promote the formation rate of C-S-H gel. With the prolongation of age, the C-S-H diffraction peak at 28 days was significantly stronger than that at 3 days, and the diffraction peak intensity of all hydration products increased.

(2)SEM-EDS analysis

Figure 6 and Figure 7 show the 3-day alkali excitation RHA-GGBS-based purification slurry specimens RHA1 and RHA3 according to scanning electron microscopy and energy spectroscopy analysis.

It can be observed from the figure that the main product of RHA1 and RHA3 paste test blocks is C-S-H gel, and its main elements include O, Ca, Si, and Al. After curing for 3 days, the hydration reaction is still in progress. There are a lot of cracks and voids in the test blocks. The relatively loose hydration products and a large amount of granular materials lead to low compactness in the early stage. It can be seen from the microstructure that 3 days’ RHA3 is more dense than RHA1. The addition of rice husk ash in RHA3 results in a high SiO_2_ content, which can react with Ca(OH)_2_ to produce a large number of C-S-H gel, making the structure more compact in the early stage, and the resulting product presents a more compact plate structure than RHA1. With the progress of hydration, hydration products C-S-H and C-A-S-H gel are continuously produced. The gel is filled in the pores between particles. In the initial stage, the hydration reaction takes place on the surface of micro-powder. In the later stage, the internal hydration of slag leads to the hydration of the silicon-rich phase. At this time, a large number of hydrolytic ions will appear to accelerate the hydration reaction products. As time goes on, a large amount of micro-powder is hydrated, and the generated gel materials are aggregated. Some are wrapped on the surface of slag particles, and some are filled in the gap of slag particles. The gap is almost filled, forming a hardened structure of dense cementitious material.

### 3.2. CPB Experimental Results and Analysis

In order to explore the optimal bone–glue ratio, we addressed the water glass modulus and alkali content for alkali excitation RHA–GGBS filling, based on an optimal dosage of 10% RHA, a fixed slurry concentration of 83%, and taking the bone–glue ratio, water glass modulus, and alkali content as variables; the mix proportion combinations of the bone gel–ratio were found to be 4, 4.5, and 5, those of the water glass modulus were 1.0, 1.2, and 1.4, and those of the alkali content were 5%, 6%, and 7%, based on three factors and three levels of design of nine groups of experiments. Table 3 shows the Orthogonal test factor level. Before the test, the cementitious material and aggregate were weighed and stirred evenly. Then, the alkali-excitant solution was configured, the weighed sodium hydroxide was poured into the beaker, and the water glass was added and stirred evenly. The water was divided into two parts, one part was placed in the beaker to stir, and the other part was rinsed with the residual solution. The mixed cementitious material and aggregate were added to the alkali excitation solution. Then, the solution was poured into a 50 mm × 100 mm mold that had been brushed with oil, and the uniaxial compressive strength of the rock triaxial testing machine was used to test its uniaxial compressive strength. The mold-loading method adopted displacement control; the specimen-loading rate was 1 mm/min, and the displacement limit was 5 mm. Table 4 shows the orthogonal mix proportion combinations.

#### 3.2.1. Test Results

The results of the orthogonal filler experiment are shown in Table 5.

#### 3.2.2. Slump Degree Analysis of CPB

The result of the extreme difference analysis of the slump of CPB is shown in Table 6. It can be seen that the difference between the bone–glue ratio, modulus, and alkali excitation content on the slump degree is 18, 49.667 and 12.333, respectively, which shows that the factor that has the greatest impact on slump is the modulus because the modulus affects the content of the total liquid, and the higher the liquid content in the slurry, the greater the slump. The variable with the second highest impact is the bone–gel ratio, which affects the amount of gelatinous material in the slurry. The variable that has the lowest impact is the alkali content, which has an impact on the alkali excitation concentration, while the speed of reactant generation also affects the slump.

(1)The effect of bone–glue ratio on slump

The larger the bone-to-glue ratio, the smaller the slump. The aggregate particles are large, the tailing sand fine aggregate particles in the CPB are fewer, and the aggregate surface is not smooth. The gelling material is also finer, and is mostly powder, with a strong water-absorption capacity, which means that the amount of cementitious material is reduced. The bone–glue ratio and modulus have the greatest impact on the filler and susceptibility; when the bone–glue is relatively large, the adhesion and water retention of the filler also worsens, and it is easy for stratified segregation and water secretion to occur.

(2)The influence of the modulus on the slump

It can be seen from the extreme difference analysis table that the modulus has the greatest influence on the slump of the alkali-activated RHA-GGBS-based filler, and the slurry fluidity is affected by the water glass modulus. As the modulus of the excitant increases, the quality of the slurry is certain, the amount of water glass increases, the water consumption is reduced, and the fluidity of the slurry is reduced. The viscosity of the water glass is determined by the SiO_2_ content, and the total solution in the CPB includes water glass, sodium hydroxide, and water, among which the proportion of water is relatively large. Because the modulus increases, the content of the water glass increases, the water glass liquid is viscous, the alkali solution becomes more viscous, and the slurry slump is reduced. With the increase in the modulus, the silicate ions in the water glass increase, which is beneficial to the disintegration of the GGBS particles. When the modulus becomes larger, more water is needed to ensure that the silicate ions can quickly form silicates with the aluminosilicon phase, the aluminosilicate plasma mass accelerates the formation of a gel, and the PH value is reduced, which is not conducive to the disintegration of the GGBS. The viscosity also increases and the fluidity becomes poor, thereby causing the slump to decrease.

(3)The influence of alkali content on the slump

When the alkali content increases, the NaOH content and alkali concentration also increase, which strengthens the alkalinity, accelerates the dissolution of the silicon aluminum phase component, promotes the formation of gel and structure recombination, and reduces the slump. The higher the alkali content, the higher the OH concentration of the excitor, thereby increasing the compressive strength of the alkali excitation GGBS purification slurry. However, there is an optimal dosage for the Na_2_O content, and excess NaOH leads to a final decrease in strength because the excess alkali content does not adequately react with the GGBS.

(4)The influence of liquid–solid ratio on the slump

The flow state of the filling slurry is not only affected by water. Water glass is liquid and obviously contains water, but the degree of influence of the two on the slump is different. Water glass is more viscous, and water has no impact on this. However, the greater the water content, the greater the dilution of the slurry, and the greater the distance between the particles, leading to a greater degree of collapse. The liquid–solid ratio has a significant impact on slump; with an increase in the liquid–solid ratio, the slump degree increases, and the larger the liquid-to-solid ratio, the more obvious the change trend. With an increase in liquid content, the inside and outside of the dry material are gradually saturated, and the remaining liquid amount can only be repelled outside, so the slump is large. When the collapse is too large, it has a negative impact on the condensation time, water excretion rate, and compressive strength of the filling. The addition of gelling material to the paste causes a great improvement. An extreme difference can therefore be seen in Table 7. The biggest factor affecting the slump is the modulus. The slump decreases with the increase in the modulus and with the increase in the bone–glue ratio, and decreases with the increase in the alkali content, before later increasing again. The larger the liquid–solid ratio, the greater the slump.

**Table 7 materials-15-04397-t007:** Analysis of range difference of compressive strength at each age.

Range Analysis	*K*	Factor *A*	Factor *B*	Factor *C*
**3 days’ compressive strength**	*K1*	1.033	1.257	1.293
	*K2*	0.917	1.030	0.887
	*K3*	0.537	0.200	0.307
	*R*	0.496	1.057	0.986
	Degree of influence	*B* > *C* > *A*		
**7 days’ compressive strength**	*K1*	5.230	5.507	2.327
	*K2*	3.840	3.887	3.110
	*K3*	1.833	1.560	5.517
	*R*	3.347	3.947	3.190
	Degree of influence	*B* > *A* > *C*		
**28 days’ compressive strength**	*K1*	6.423	6.760	2.547
	*K2*	4.633	4.650	4.333
	*K3*	2.440	2.087	6.617
	*R*	3.983	4.673	4.070
	Degree of influence	*B* > *C* > *A*		

#### 3.2.3. Filling Strength Analysis

The UCS range analysis of the CPB is shown in Table 7.

(1)The effect of bone glue–ratio on strength

As the bone-to-gel ratio increases, the compressive strength of the filler decreases. The pores in the filler become larger, and the number of pores increases. The increase in the bone-to-glue ratio means that the proportion of the active ingredient of the cementitious material in the filling system is reduced, resulting in a decrease in the hydration reaction rate and an insufficient hydration reaction. The tailing sand particles in the filling cannot be completely enveloped by the gel, which gathers together in large quantities, and the gel system does not form a dense network structure. As a result, the supporting role played by the skeleton is not strong, resulting in a decrease in the strength of the macroscopic filling. The particle size of the aggregate is large and the surface is rough, which makes the porosity of the CPB relatively large.

(2)The influence of water glass modulus on strength

As the water glass modulus increases, the compressive strength shows a downward trend. The water glass modulus increases, the NaOH content decreases, the content of NaOH determines the alkaline strength of the solution. In contrast, when the modulus becomes larger, the OH-ion concentration becomes smaller, the silicate ion concentration provided by the water glass is too high, and the silicate ion itself is polymerized, which is not conducive to the ion reaction and causes the strength to decrease. If the water glass modulus is too small, although the Na_2_O content is high, it is beneficial to the disintegration of the vitreous body, but the silicate ions are few and can only rely on Na_2_O, so it is not conducive to the development of strength. NaOH promotes GGBS disintegration, and silica gel in water glass is formed with Ca^2+^ and Al^3+^ to produce C-(A)-S-H gels. The water glass modulus of the excitant affects the silicon-oxygen tetrahedron, the low water glass modulus has a low degree of polymerization with the silicon-oxygen tetrahedron, the number of monomers makes the activity high, the alkali excitation reaction is fast, and the compressive strength increases.

(3)The effect of alkali content on strength

The increase in the alkali content enhances the alkalinity of the stimulator, which is conducive to stimulating the hydration rate of the cementitious material. When the alkali content is low, the gelling system produces fewer hydration products so that the filling cannot form a dense structure, meaning that the compressive strength of the filling is low.

The extreme difference analysis of the unconfined compressive strength of the filler at each age is shown in Table 8. The compressive strength exemplifies the basic performance of the filler and one of the decisive factors regarding its advantages and disadvantages. It can be seen from the table that the primary and secondary relationships affecting the compressive strength of the CPB at 3 days and 28 days are the water glass modulus > alkali content > bone–glue ratio, while the primary and secondary relationships of the compressive strength of the CPB at 7 days are the water glass modulus > bone–glue ratio > alkali content. Among them, the water glass module is the main factor, and the polarity at 3 days, 7 days, and 28 days reaches 1.057, 3.947, and 4.673, respectively. The alkali content is a secondary influencing factor, and the range also reaches 0.986, 3.190, and 4.070, respectively. According to the water glass modulus, bone–gel ratio, and alkali content, the 3-day optimal scheme is determined to be A1, B1, C1, the 7-day optimal scheme is A1, B1, C3, and the 28-day optimal scheme is A1, B1, C3.

The influence factors of 3-day unconfined compressive strength were the bone–glue ratio and sodium silicate modulus in inverse proportion to compressive strength; alkali content increased, and compressive strength increased after decreasing. The unconfined compressive strength at 7 days and 28 days was inversely proportional to the bone–glue ratio and sodium silicate modulus, and the alkali content was positively proportional. The fifth group had the highest intensity, in which the first and seventh groups had a higher intensity. When the modulus was 1.0, the excitation effect of the cementitious material was better. When the modulus of sodium silicate is low, the content of NaOH is higher, and the alkalinity is stronger, which is more favorable to the promotion of a hydration reaction, so the reaction of the product is faster, and the strength is higher.

#### 3.2.4. Optimal Mix Ratio Analysis

According to the experimental analysis, consistent with the slump requirements, the fourth and seventh groups did not meet the requirements due to excessive collapse, and the ninth group did not meet the requirements because the collapse was too small. From the compressive strength analysis table, it can be seen that the bone-to-glue ratio is 4:0, the modulus is 1.0, and the compressive strength is high. The alkali excitor content is better in high strength, but considering the cost and actual strength, Scheme 1 satisfies the slump requirements and the compressive strength is higher, so Scheme 1 is the optimal mix ratio.

#### 3.2.5. SEM-EDS Analysis

In this experiment, the internal structure and morphology of the hydration products of the CPB were observed by SEM and EDS methods, and the composition of the products was analyzed.

First, a small sample of the net slurry that had reached the test age was taken, the carbonized surface was scraped off, and it was placed in absolute ethanol and soaked until the reaction was no longer carried out. It was dried before testing, fixed, then treated with gold spray, and placed under a scanning electron microscope for observation.

By optimally analyzing CPB Group 1 as the optimal mix ratio, the test block of the prepared filling material at 28 days was selected for microstructural analysis. Its microstructure was detected, the material was micro-scanned using a Hitachi S4800 scanning electron microscope, and some small test blocks in the CPB were selected from the microstructure analysis sample. The SEM observations are shown in Figure 8, and the EDS energy spectrum analysis is shown in Figure 9.

The microstructure of the filler contains a small number of pores, voids, and cracks. The void may have been left after the water filled the existing pores and evaporated. The cracks may be shrinkage cracks caused by curing, or they may be shrinkage compression loads caused by curing. In either case, the presence of voids and cracks in the filler will lead to a decrease in the strength of the filler.

From the microscopic diagram, a large number of flocculent and small particles of gelatinous substances can be seen, and oxygen, silicon, aluminum, and calcium are also used as the main elements in the EDS, which also indicates the presence of C-A-S-H gels and C-S-H gels in the reaction products. Quartz is a silicon-based oxide, and the presence of the gel is also the main influencing factor of the filling strength mechanism. The coexistence of this C-S-H gel and C-A-S-H gel regulates strength and permeability. The alkali excitation of the curing age of 28-day RHA–GGBS-based filler SEM is shown in Figure 8; EDS point sweep elemental analysis is shown in Figure 9.

(1)White flocculent: In the lower right corner of Figure 8a, a white mass flocculent product appears, and the EDS energy spectroscopy in Figure 9b shows that the levels of O, Si, and Ca are high, so the substance is mainly a C-S-H gel, and the C-S-H gel comprises the Ca(OH)_2_ co-reacted using the active ingredient SiO_2_ in RHA and the hydration reaction in the GGBS. The reaction Equation is [38]:SiO_2_ + Ca(OH)_2_ + H_2_O→CaO·SiO_2_·xH_2_O(1)

(2)Small round grain cluster aggregates: In the upper right corner of Figure 8a, there are some relatively concentrated and small spherical cluster-like substances. The content analysis of element in Figure 9c found that O, Si, Al, and Ca account for a relatively large proportion, and the material is analyzed as a calcium hydrated silica aluminate gel. The following reactions occur with this gel [39]:Al_2_O_3_ + Ca(OH)_2_ + 2SiO_2_ + 3H_2_O→CaO·Al_2_O_3_·2SiO_2_·4H_2_O(2)

(3)Circular agglomerate clusters: The EDS in Figure 9a contains the main elements, oxygen, silicon, aluminum, magnesium, and calcium, and the main hydration products can be analyzed as C-S-H and hydrotal rock (Mg_6_Al_2_CO_3_(OH)_16_·4(H_2_O)).

## 4. Conclusions and Prospects

(1)Through the experimental study of the alkali-activated RHA-GGBS-based purification solution, taking slump, setting time, and uniaxial compression test as indicators, it is determined that the slump decreases with the increase of RHA content. According to the change of RHA under different substitution rates, the optimal substitution rate of RHA is determined to be 10%.(2)The activity of the cementitious material is stimulated by the alkali excitor, and the RHA interacts with the GGBS to generate C-(A)-S-H gel, which is the main mechanism of compressive strength formation. With the increase in the amount of RHA used in the mixture, the compressive strength first increases and then decreases. As the amount of RHA blending is gradually increased from 5% to 10%, the internal structure becomes denser, and the amount of C-S-H gel formed becomes greater. When 15% of the mixture is composed of RHA, the progress of the reaction is affected, thereby reducing the intensity.(3)In the microscopic analysis of alkali excitation in the GGBS-based pure slurry and RHA-GGBS-based pure slurry, the internal structure of RHA3 when added to the RHA is relatively dense, and the resulting C-S-H gel therefore becomes denser. The gel and RHA fill in its pores, meaning that its strength increases, but the microstructure still contains microcracks.(4)Based on the orthogonal design method, the experiment is optimized by taking the bone–gel ratio, modulus, and alkali content as the variable factors and the strength and slump degree as the targets. Through an analysis of the extreme difference and variance in the bone–glue ratio, modulus, and alkali content in the test results, a bone–gel ratio of 4:0, a water glass modulus of 1.0, and an alkali content of 5% was found to be the optimal mix ratio. Of these variables, the water glass module has the greatest impact on both slump and compressive strength. Through microscopic analysis, the main reactants were found to be calcium silicate gel, hydrotalcline, and calcium aluminum aluminate gel hydrate.

There are two main innovations in the research of RHA-GGBS-based cementitious material: (1) The RHA-GGBS CPB has been deeply studied for the first time. Through the microscopic analysis and the influence analysis of fluidity, it is proposed that RHA plays a key role in the compressive strength of the paste. (2) The filling preparation and mechanical properties of RHA-based cementitious material were studied for the first time. Through orthogonal experiments, the optimum mix proportion of alkali-activated RHA-GGBS CPB was obtained.

In this paper, the preparation methods of RHA-GGBS-based materials were studied, and a series of influencing factors were analyzed. The paper also has certain limitations, and we can further study the topic through more tests. The following aspects need to be further explored: in this paper, the durability test of alkali-activated RHA-GGBS backfill can be carried out to study the durability issues under different solution erosions, freeze–thaw cycles, and coupling states. 

## Figures and Tables

**Figure 1 materials-15-04397-f001:**
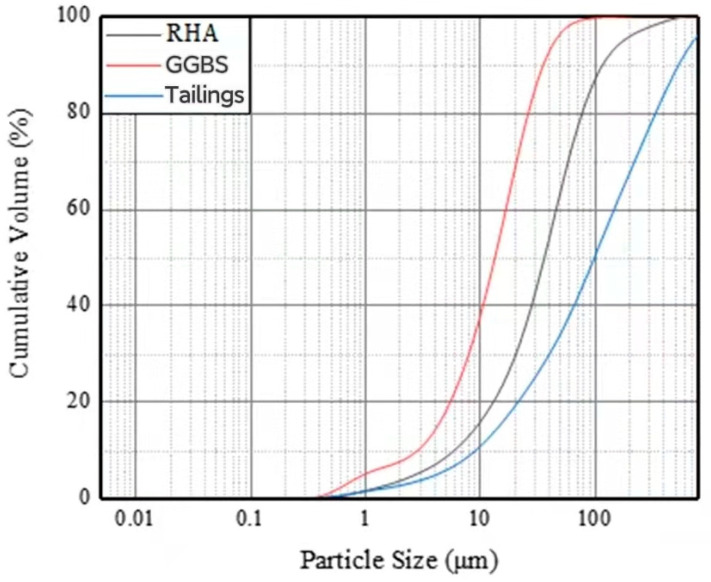
Particle size curve of cementitious material.

**Figure 2 materials-15-04397-f002:**
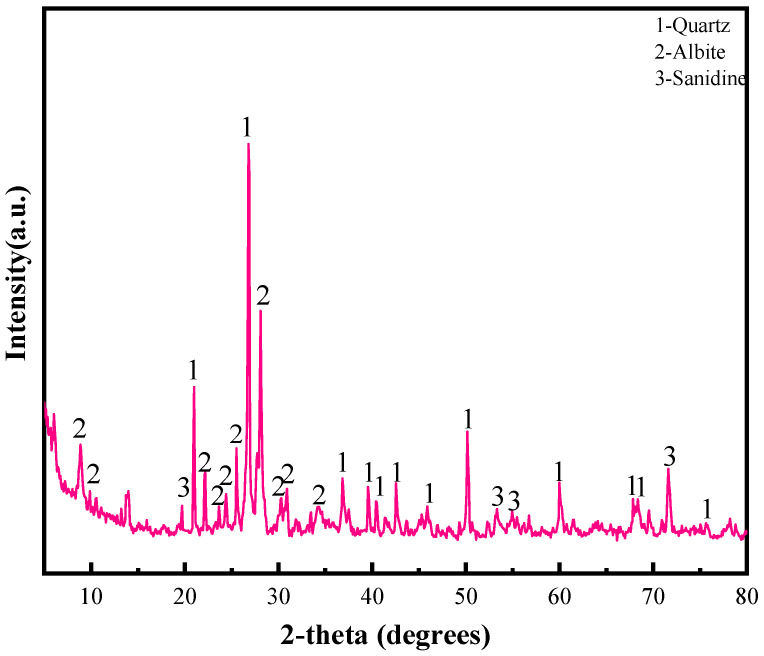
XRD of tailing sand.

**Figure 4 materials-15-04397-f004:**
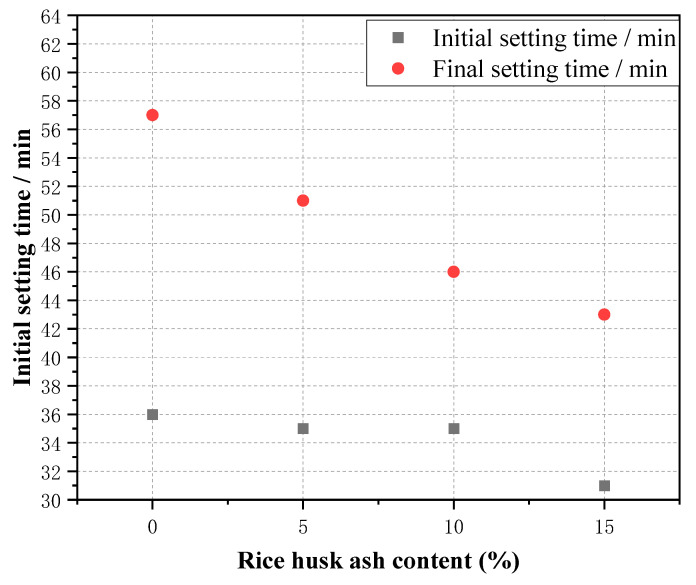
Setting time of purified slurry.

**Figure 5 materials-15-04397-f005:**
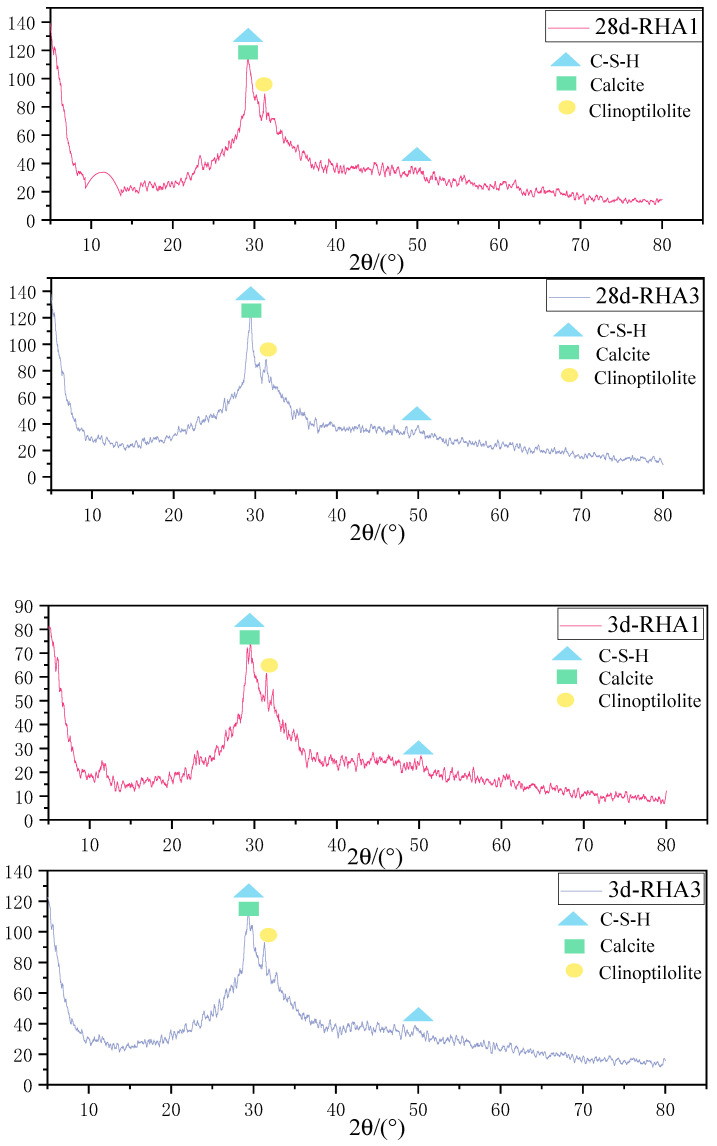
XRD analysis of alkali-activated GGBS slurries with different amount of RHA at 3 days and 28 days.

**Figure 6 materials-15-04397-f006:**
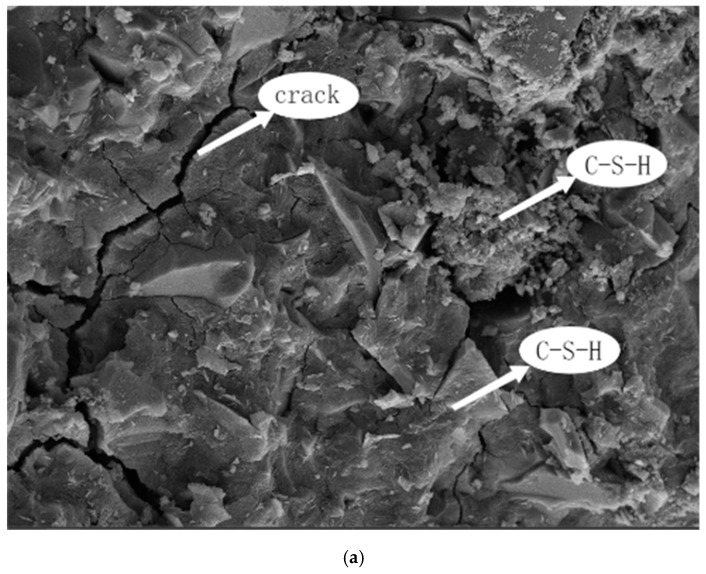
RHA1 microgram. (**a**) Micromorphology of RHA1. (**b**) Upper right arrow in figure. (**c**) Lower right arrow in figure.

**Figure 7 materials-15-04397-f007:**
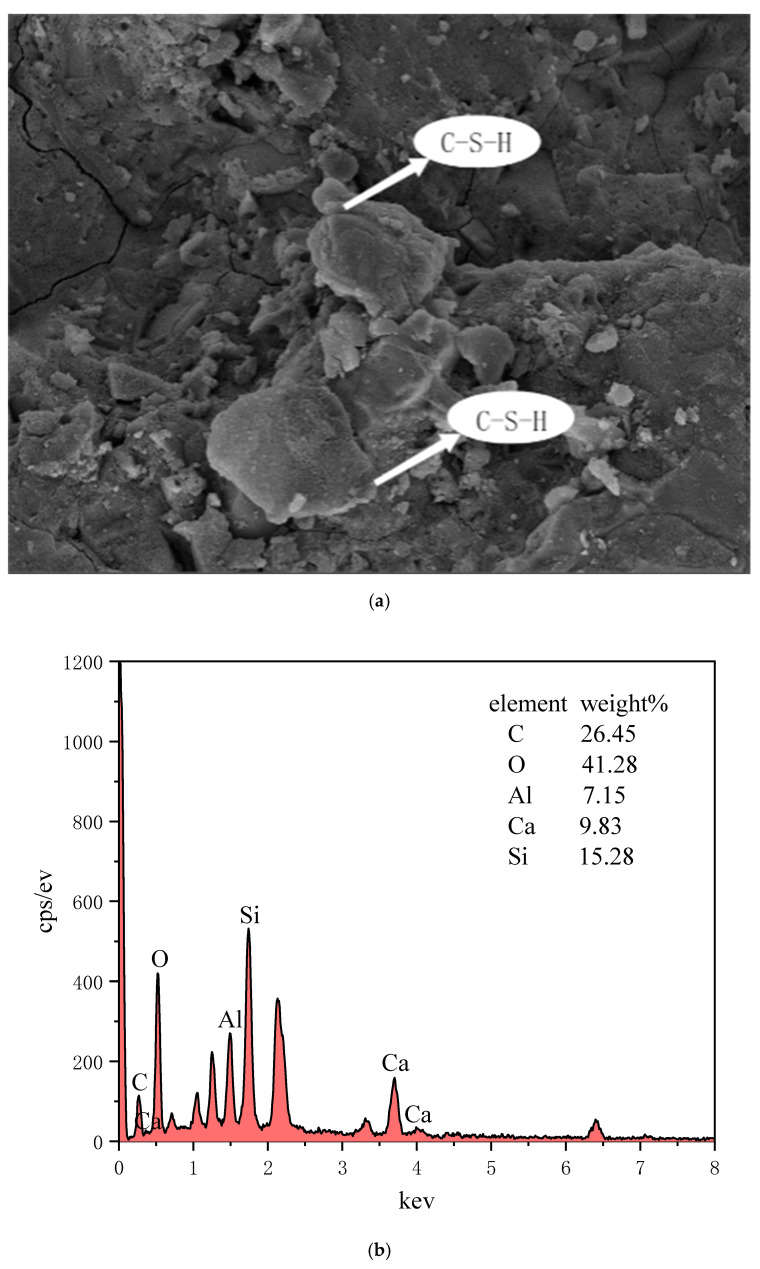
RHA3 microgram. (**a**) Micromorphology of RHA3. (**b**) Upper arrow in figure. (**c**) Lower arrow in figure.

**Figure 8 materials-15-04397-f008:**
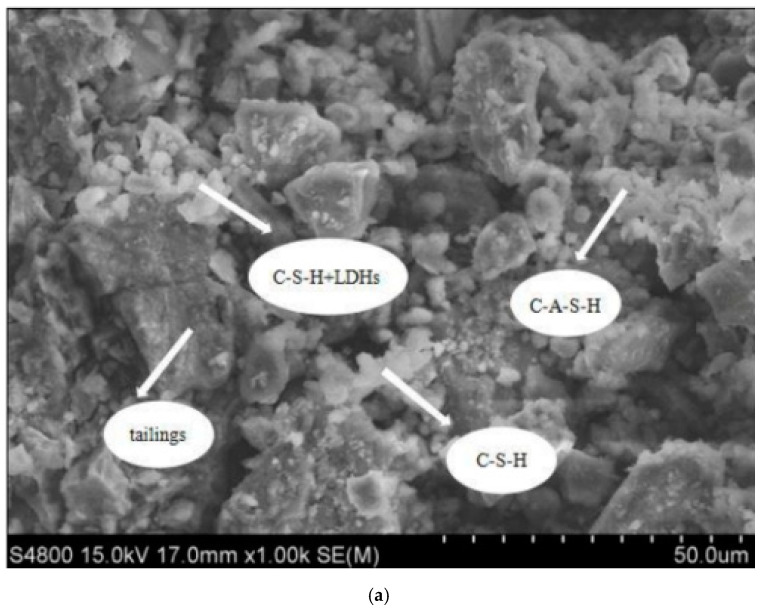
Micromorphology of CPB. (**a**) Magnified 200 times and (**b**) Magnified 50 times.

**Figure 9 materials-15-04397-f009:**
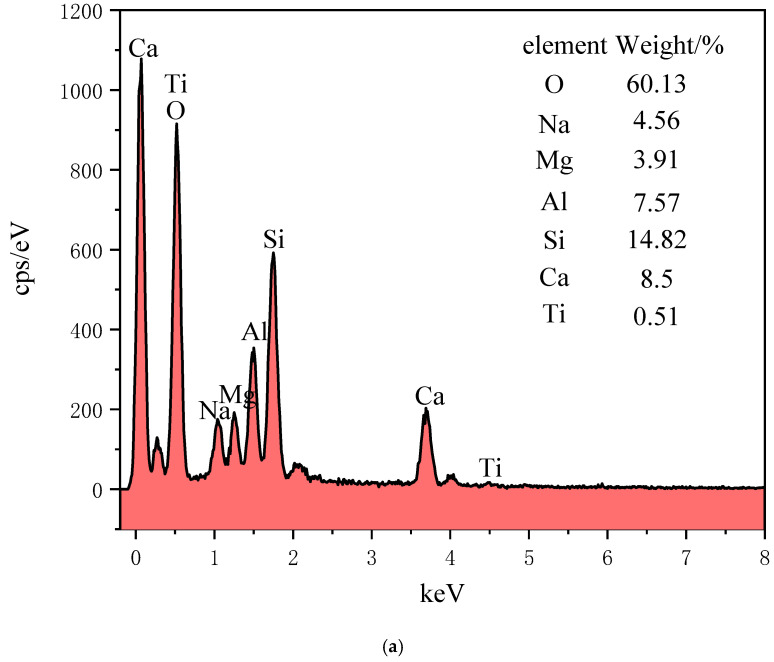
EDS point sweep. (**a**) Upper left corner substance in Figure 8a, (**b**) Lower left corner substance in Figure 8a and (**c**) Upper left corner substance in Figure 8a.

**Table 1 materials-15-04397-t001:** Chemical composition of raw materials (wt%).

Raw Materials	CaO	Fe_2_O_3_	SiO_2_	Al_2_O_3_	MgO	K_2_O	SO_3_
RHA	2.63	1.63	83.62	3.01	0.96	4.59	0.89
GGBS	34.15	0.20	31.14	19.98	10.49	0.29	2.07
Tailing	4.45	16.30	45.39	14.94	5.47	3.86	0.61

**Table 2 materials-15-04397-t002:** Mix proportion of clean slurry.

Group	RHA Content (%)	GGBS Content (%)	Modulus	Alkali Consumption (%)	*W/B*
RHA1	0	100	1.0	4	0.35
RHA2	5	95	1.0	4	0.35
RHA3	10	90	1.0	4	0.35
RHA4	15	85	1.0	4	0.35

**Table 3 materials-15-04397-t003:** Orthogonal test factor level table.

Factor Level	Bone–Glue Ratio *A*	Water Glass Modulus *B*	Alkali Excitation Percentage *C* (%)
1	4	1.0	5
2	4.5	1.2	6
3	5	1.4	7

**Table 4 materials-15-04397-t004:** Orthogonal mix proportion combinations.

Test Number	Bone–Glue Ratio *A*	Water Glass Modulus *B*	Alkali Excitation Percentage *C* (%)
1	4	1.0	5
2	4	1.2	6
3	4	1.4	7
4	4.5	1.0	6
5	4.5	1.2	7
6	4.5	1.4	5
7	5	1.0	7
8	5	1.2	5
9	5	1.4	6

**Table 5 materials-15-04397-t005:** Orthogonal test results.

Test Number	Slump /mm	3 Days Compressive Strength/MPa	7 Days Compressive Strength/MPa	28 Days Compressive Strength/MPa
1	253	1.76	6.65	7.14
2	245	1.02	4.73	6.53
3	230	0.32	4.31	5.60
4	275	0.55	4.46	6.20
5	206	2.02	6.83	7.31
6	239	0.18	0.23	0.39
7	263	1.46	5.41	6.94
8	238	0.05	0.10	0.11
9	173	0.10	0.14	0.27

**Table 6 materials-15-04397-t006:** Collapse range analysis table.

Range Analysis	*K*	Factor *A*	Factor *B*	Factor *C*
**Slump**	*K1*	242.667	263.667	243.333
	*K2*	240	229.667	231
	*K3*	224.667	214	233
	*R*	18	49.667	12.333
	Degree of influence	*B* > *A* > *C*		

**Table 8 materials-15-04397-t008:** Table of variance analysis of compressive strength.

Variance Analysis	Factor	Sum of Squares of Deviations	Freedom	*F* Ratio	Significant Level
**3 days’ compressive strength**	Bone–glue ratio	0.405	2	1.000	3
	Modulus	1.857	2	4.585	1
	Alkali content	0.879	2	2.170	2
**7 days’ compressive strength**	Bone–glue ratio	16.961	2	0.836	2
	Modulus	23.614	2	1.164	1
	Alkali content	16.582	2	0.817	3
**28 days’ compressive strength**	Bone–glue ratio	23.882	2	0.842	3
	Modulus	32.863	2	1.158	1
	Alkali content	24.971	2	0.880	2

## Data Availability

Date can be obtained from corresponding authors upon reasonable request.

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
