# Peer review of "Preparation and Microstructure of Alkali-Activated Rice Husk Ash-Granulated Blast Furnace Slag Tailing Composite Cemented Paste Backfill"

_materials, 2022, doi:10.3390/ma15134397_

Round 1

Reviewer 1 Report

The work "Preparation and microstructure of alkali-activated rice husk ash-granulated blast furnace slag tailing composite cemented paste backfill" has been reviewed; 

1. RHA has been used as alternative cement in different areas of the construction works; it would be qualitative to include the pozzolanic strength, three chemical moduli and the reactivity rate of this ash material. 

2. The RHA has been observed to improve workability of concrete or pastes thereby reducing the water/binder ratio and consequently improving strength usually affected by high water/binder ratio, what is the orthogonal quality of the RHA that helps it occupy this property?

3. GGBS is richer in CaO content but cementitious still, what is the main purpose of using RHA and GGBS in the same when they possess the same cementing properties?

4. The mix proportions prepared to produce the 9 outcomes of the strength properties of the paste were not listed in the paper and that seems to be an aberration. Your reader may want to know the interesting 9 mixes that gave rise to the 9 outcomes of this paper. 

5. With slump ranging between 173 and 275, this should be considered a self compacting concrete, please justify!

6. Strength of up to 10MPa was not produced by the concrete, why? There is a serious need to include the 9 concrete mixes that gave rise to these 9 outcomes for a reader to justify why the concrete studied produced low strength. Is this a lightweight concrete?

7. I am wonderful if there was something missing in page 13 where it is reported that there exist large pores in a concrete with RHA and GGBS and also tailing sand of fine texture. Please justify and revise where necessary.

Reviewer 2 Report

General comments on the text as a whole:

- Please provide details of the captions under the figures and the titles of the tables. The captions in the article are too laconic. For example, there is missing information about the labeling of the test samples in Figures 8 and 9.

 - Please consider and improve the literature citations throughout Chapter 1 Introduction. In my opinion, for example, the notation on page 2 "Zain M F M et al." is not very fortunate. Maybe it is worth using the notation used in the references list - "Zain, M.F.M., et al."

- Attention when writing references. They are not unitary. Plesae use MDPI Reference List and Citations Style Guide, especially when recording publications that have multiple authors. Please include the names of all authors, not just the first author.

Detailed comments:

Page 2 – please explain the abbreviation SCM, when first mentioned, abbreviations should be explained

Page 2 – please explain the abbreviation PP, when first mentioned, abbreviations should be explained

Page 3 – please explain the abbreviation SF, when first mentioned, abbreviations should be explained

Page 5 – the following statement was used: „… S95 GGBS micronized powder, produced by Shandong Kangjing New Material Technology Co., Ltd.” – please characterize this material - state the source of origin, from which production process (e.g. from ferrous or non-ferrous metallurgy).  Moreover, please provide the characteristics of the tested GGBS waste - once it is mentioned micronized powder (page 5) and on page 6 in, caption under figure 2 it is mentioned tailing sand.

Page 7, section 3.1.1 - please correct "295mm" to "295mm", likewise three other examples on this page. Please also correct "24.5 cm" to "24.5 cm".

Page 8 - please correct "33.89 MPa", "38.45 MPa" and "36.71 MPa" to 33.89 MPa, 38.45 MPa and 36.71 MPa respectively

Pages 10-11 – Figures 4 and 5 – Please explain the differences in the course of XRD curves for sample 28d-RHA1, which is placed both in Fig. 4 and Fig. 5. The shape of the peaks for C-S-H, calcite and clinoptilolite is different in these curves. There is no information as to why the diffraction patterns are only presented for the RHA1 and RHA3 samples. If these are examples of diffraction patterns, please underline (emphasize) this in the text.

Page 18 - A citation for figure 10 appears on this page. The article contains nine figures, so please check the numbering of the figures.

Reviewer 3 Report

Ms. Ref. No.: materials-1758748 – peer-review

Preparation and microstructure of alkali-activated rice husk ash-granulated blast furnace slag tailing composite cemented paste backfill

Reviewer comments:

SUMMARY

The manuscript is a paper related to alkali-activated rice husk ash-granulated blast furnace slag. This is a topic that has not been widely covered in the literature, therefore, this a subject of great interest, but it is somehow limited in the analysis and application of these results.

MAIN IMPRESSIONS

This paper has an undeniable practical usefulness. However, from a scientific point of view, the following issues must be addressed: i) Reviewed findings should be discussed in deep, ii) the novelty of the paper should be underlined and iii) it is recommended to underscore the novelty of this paper in the conclusion.

MORE DETAILED COMMENTS

The authors did not follow the mdpi template:

·         The lines are nor numbered. Therefore, the review is a little bit more complicated.

·         References: None of them have DOI.

·         References did not follow the mdpi rules.

Page 1: Define the CPB acronym the first time that you have used it.

Page 1: “alkali to excite rice husk ash…”.  Probably is better “Alkali-activation”.

Excite: https://dictionary.cambridge.org/es/diccionario/ingles/excite

·         to make someone have strong feelings of happiness and enthusiasm:

·         to cause a particular reaction in someone:

·         to make something, for example particles or cells, more active:

Alkali-activation is a globally growing technology that involves the chemical reaction between a solid aluminosilicate precursor and an alkaline activator, at room temperatures, giving a hardened product (Shi et al. 2006).

Page 2:   “.. found that rice husk ash (RHA) is rich in silica and, with appropriate combustion techniques, can be produced from rice husks for use in concrete …”. This fact was found many years ago. For instance: Salas J, Castillo P, Sanchez Rojas MI, Veras J (1986) Use  or  rice  husk ash  an  addition   in  mortar  Mater Construcc 36:21–39. https://materconstrucc.revistas.csic.es/index.php/materconstrucc/article/view/888/1205

 Sánchez de Rojas, M. I., Frías, M., & Rivera, J. (2000). Studies about the heat of hydration developed in mortars with natural and by-product materials. Materiales De Construcción, 50(260), 39–48. https://doi.org/10.3989/mc.2000.v50.i260.389

Page 2:  auxiliary cementitious material” or “supplementary cementitious material.”?

Page 2: “… high carbonization resistance,..” or “… high carbonation resistance,..”?

Carbonization is the process of changing or being changed into carbon, by burning, heating, or during fossilization.

Page 2:  “…and an acidic environment,” or “…and an acidic environment resistance,”?

All the manuscript: The authors should check the technical English throughout the manuscript. Apparently, the manuscript has been checked by MDPI(www.MDPI.com ---    for editing the English language – page 19).

Page 2:  Could you please add a reference?: “…in the rate of RHA substitution.

…”

Page 2:  Could you please change “water-to-glue ratios” to water-to-binder ratio”?

Glue: a sticky substance that is used for joining things together permanently, produced from animal bones and skins or by a chemical process (https://dictionary.cambridge.org/es/diccionario/ingles/glue)

Pages 7-18:

Introduction: There are 33 references. However, “3. Results and Discussion” does not use any refence to discuss the results.

Figures 1-9 are very small. Letters should be bigger.

Page 5: Table 1: Could you please add the Na2O content?

Page 5: Table 1: Could you please add the P2O5 content for the tailing?

Page 8: “…the compressive strength at each age was better than that of the other groups. ..”. Could you please explain why it was better than that of the other groups? Could you please use the references given in the introduction to explain it?

Page 9: Have you found any correlation between flow and compressive strength?

Page 19: Author Contributions is wrong. Could you please follow the mdpi template?

RECOMMENDATION

In conclusion, Major changes have been proposed.

Reviewer 4 Report

Preparation and microstructure of alkali-activated rice husk ash-granulated blast furnace slag tailing composite cemented paste backfill

 Manuscript Number:1758748

In the present paper authors have experimentally investigated alkali-activated rice husk ash and granulated blast furnace slag to prepare cementitious material. The influence of rice husk ash dosage on the mechanical properties of alkali-activated rice husk ash-granulated blast furnace slag tailing composite cemented paste backfill. The outcome of this research is significant in providing useful information about mechanical behavior of alkali-activated materials. However, the paper requires major improvement before it can be recommended for publication, it is proposed to re-submit a thoroughly revised version of the manuscript, considering the following comments and they are listed in the following (not in order of their importance):

1.     Title and abstract are ok.
2.      Overall recommendation should be reported in one sentence at the end of the abstract
3.     The authors should overview the recent progress made in the relevant area in the past two years or so. Such as: https://doi.org/10.3390/min8090381; https://doi.org/10.1016/j.jclepro.2019.118568; https://doi.org/10.12989/acc.2021.11.2.127
4.     The first paragraph of introduction section is very small. The reader can be paid attention after reading the first paragraph. Authors may provide the background of the study, research significance, research questions solved/novelty in this study and application of geopolymer.
5.     Please explain in detail the chemical and physical properties that affect fluidity.

  1. In SEM-EDS analysis section,  the sentence structure needs to be modified, and the analysis and interpretation of the content are too rough.
  2. Table 8 and whole manuscript. Instead of 3D, write it 3 days.

8.     The conclusions need to be improved by making them more effective. Also highlight the assumptions and limitations.

9.     Conclusion: Can authors highlight future research directions and recommendation?

Round 2

Reviewer 1 Report

The authors have sufficiently done the required revisions

Reviewer 3 Report

Accept the paper